# Development and Assessment of a Pooled Serum as Candidate Standard to Measure Influenza A Virus Group 1 Hemagglutinin Stalk-Reactive Antibodies

**DOI:** 10.3390/vaccines8040666

**Published:** 2020-11-09

**Authors:** Juan Manuel Carreño, Jacqueline U. McDonald, Tara Hurst, Peter Rigsby, Eleanor Atkinson, Lethia Charles, Raffael Nachbagauer, Mohammad Amin Behzadi, Shirin Strohmeier, Lynda Coughlan, Teresa Aydillo, Boerries Brandenburg, Adolfo García-Sastre, Krisztian Kaszas, Min Z. Levine, Alessandro Manenti, Adrian B. McDermott, Emanuele Montomoli, Leacky Muchene, Sandeep R. Narpala, Ranawaka A. P. M. Perera, Nadine C. Salisch, Sophie A. Valkenburg, Fan Zhou, Othmar G. Engelhardt, Florian Krammer

**Affiliations:** 1Department of Microbiology, Icahn School of Medicine at Mount Sinai, One Gustave L. Levy Place, Box 1124, New York, NY 10029, USA; jm.carreno@mssm.edu (J.M.C.); Raffael.Nachbagauer@modernatx.com (R.N.); mabehzadi1985@gmail.com (M.A.B.); shirin.strohmeier@mssm.edu (S.S.); lynda.coughlan@mssm.edu (L.C.); teresa.aydillo-gomez@mssm.edu (T.A.); Adolfo.Garcia-Sastre@mssm.edu (A.G.-S.); 2Division of Virology, National Institute for Biological Standards and Control (NIBSC), South Mimms, Potters Bar EN6 3QG, UK; Jacqueline.McDonald@nibsc.org (J.U.M.); tara.hurst@bcu.ac.uk (T.H.); Lethia.Charles@nibsc.org (L.C.); 3Division of Analytical and Biological Sciences, National Institute for Biological Standards and Control (NIBSC), South Mimms, Potters Bar EN6 3QG, UK; Peter.Rigsby@nibsc.org (P.R.); Eleanor.Atkinson@nibsc.org (E.A.); 4Department of Biotechnology, University of Natural Resources and Life Sciences, 1190 Vienna, Austria; 5Global Health and Emerging Pathogens Institute, One Gustave L. Levy Place, Box 1124, New York, NY 10029, USA; 6Janssen Vaccines & Prevention BV, 2333 CP Leiden, The Netherlands; BBrande1@its.jnj.com (B.B.); kkaszas@ITS.JNJ.com (K.K.); lmuchen@ITS.JNJ.com (L.M.); NSalisch@its.jnj.com (N.C.S.); 7Department of Medicine, Division of Infectious Diseases, Icahn School of Medicine at Mount Sinai, One Gustave L. Levy Place, Box 1124, New York, NY 10029, USA; 8The Tisch Cancer Institute, Icahn School of Medicine at Mount Sinai, One Gustave L. Levy Place, Box 1124, New York, NY 10029, USA; 9Influenza Division, Centers for Disease Control and Prevention, Atlanta, GA 30329, USA; mwl2@cdc.gov; 10VisMederi Research srl, 53100 Siena, Italy; alessandro.manenti@vismederiresearch.com; 11Vaccine Immunology Program (VIP), Vaccine Research Center (VRC), National Institutes of Allergy and Infectious Diseases (NIAID), National Institutes of Health (NIH), Bethesda, MD 20892, USA; adrian.mcdermott@nih.gov (A.B.M.); sandeep.narpala@nih.gov (S.R.N.); 12Department of Molecular and Developmental Medicine, University of Siena, 53100 Siena, Italy; montomoli@unisi.it; 13School of Public Health, LKS Faculty of Medicine, The University of Hong Kong, Hong Kong, China; mahenp@hku.hk (R.A.P.M.P.); sophiev@hku.hk (S.A.V.); 14Influenza Center, Department of Clinical Science, University of Bergen, 5021 Bergen, Norway; fan.zhou@uib.no; 15K.G. Jebsen Center for influenza vaccines, Department of Clinical Science, University of Bergen, 5021 Bergen, Norway

**Keywords:** influenza vaccine, serology, hemagglutinin, stalk, standardization

## Abstract

The stalk domain of the hemagglutinin has been identified as a target for induction of protective antibody responses due to its high degree of conservation among numerous influenza subtypes and strains. However, current assays to measure stalk-based immunity are not standardized. Hence, harmonization of assay readouts would help to compare experiments conducted in different laboratories and increase confidence in results. Here, serum samples from healthy individuals (*n* = 110) were screened using a chimeric cH6/1 hemagglutinin enzyme-linked immunosorbent assay (ELISA) that measures stalk-reactive antibodies. We identified samples with moderate to high IgG anti-stalk antibody levels. Likewise, screening of the samples using the mini-hemagglutinin (HA) headless construct #4900 and analysis of the correlation between the two assays confirmed the presence and specificity of anti-stalk antibodies. Additionally, samples were characterized by a cH6/1N5 virus-based neutralization assay, an antibody-dependent cell-mediated cytotoxicity (ADCC) assay, and competition ELISAs, using the stalk-reactive monoclonal antibodies KB2 (mouse) and CR9114 (human). A “pooled serum” (PS) consisting of a mixture of selected serum samples was generated. The PS exhibited high levels of stalk-reactive antibodies, had a cH6/1N5-based neutralization titer of 320, and contained high levels of stalk-specific antibodies with ADCC activity. The PS, along with blinded samples of varying anti-stalk antibody titers, was distributed to multiple collaborators worldwide in a pilot collaborative study. The samples were subjected to different assays available in the different laboratories, to measure either binding or functional properties of the stalk-reactive antibodies contained in the serum. Results from binding and neutralization assays were analyzed to determine whether use of the PS as a standard could lead to better agreement between laboratories. The work presented here points the way towards the development of a serum standard for antibodies to the HA stalk domain of phylogenetic group 1.

## 1. Introduction

As defined in the strategic plan from the National Institute of Allergy and Infectious Diseases [1], some of the key points to achieve the development of effective Universal Influenza Vaccines (UIV) include: the characterization of the immune responses elicited during influenza virus infection and vaccination; establishment of novel non-hemagglutination inhibition (HAI) correlates of protection; rational design of antigens with a wider breadth of protection; and implementation of these candidates in phase I-II clinical studies. Many current efforts towards the development of these novel types of vaccines rely on the induction of effective long-term antibody responses against conserved regions of the influenza virus glycoproteins [2].

The stalk domain of the hemagglutinin (HA) has been identified as a suitable target for universal influenza virus vaccines due to its unique properties. Contrary to the head domain, which is highly plastic [3], the stalk domain exhibits a high degree of conservation among numerous influenza virus subtypes and strains [4,5,6,7] but is immuno-subdominant [8,9]. As reviewed [10], anti-stalk antibodies act through diverse mechanisms including blocking the fusion of viral and cellular membranes [11,12,13], impeding the release of viral particles from infected cells [7,14], blocking the cleavage of the hemagglutinin [5], inducing complement activation [15] and triggering FcR-mediated effector functions, namely antibody-dependent cellular cytotoxicity (ADCC) and antibody-dependent cellular phagocytosis (ADCP) [16,17]. Importantly, there is extensive evidence of the protective potential of anti-stalk antibodies in diverse animal models [6,7,14,16,18,19,20,21] and in humans [22,23,24,25]. Moreover, several vaccine candidates targeting this domain are in late pre-clinical, or early clinical stages of development [2,18,19,26,27].

Given the importance of qualitatively and quantitatively detecting antibody responses against the stalk in current research settings, and likely in future prophylactic scenarios for universal influenza virus vaccines, we initiated a collaborative project to investigate the possibility of developing an international standard serum to measure group 1 HA stalk-reactive antibodies (group 1: H1, H2, H5, H6, H8, H9, H11, H12, H13, H16, H17 and H18). As stated by the World Health Organization (WHO), ‘reference standards are used as calibrators in assays’ and define an internationally agreed, arbitrary unit that allows comparison of biological measurements worldwide [28]. There is a wide repertoire of WHO standards available, including 22 biological reference preparations in the “immunoglobulins and human sera” category and 83 biological reference preparations in the “vaccines/toxoids/toxins” category, with only two international standards available for influenza virus research [29]: (1) a standard, established in 2008, consisting of a pooled polyclonal serum obtained from individuals vaccinated with a clade 1 H5N1 virus (A/Vietnam/1194/2004) derived vaccine [30], and (2) the second International Standard for antibodies to pandemic H1N1 virus, consisting of pooled plasma from individuals who received a pandemic H1N1 split vaccine produced from the reassortant virus NYMC X-179A, derived from A/California/07/2009 [31]. Both standards were characterized by hemagglutination inhibition (HI) and virus neutralization (MN) assays. However, none of the available influenza antibody-standards are specific against the stalk of the HA. Therefore, the development of an international serum standard to measure stalk-reactive antibodies would have important implications worldwide, because it would contribute to the harmonization of assay read-outs, hence facilitating the comparison of experiments conducted in different laboratories and increasing confidence in results.

## 2. Materials and Methods

**Cells, viruses, proteins and sera.** Cells were maintained in Dulbecco’s modified Eagle’s medium (DMEM; Gibco, NY, USA), supplemented with 10% fetal bovine serum (FBS; HyClone, MA, USA) and penicillin (100 U/mL)-streptomycin (100 μg/mL) solution (Gibco, NY, USA). Madin–Darby canine kidney cells (MDCK) were used for neutralization assays; MDCK cells expressing the protein cH6/1 (cH6/1-MDCK) (Chromikova et al., 2020), which contains the exotic avian HA head domain (H6) from an H6N1 virus (A/Mallard/Sweden/81/2002) and the stalk (H1) from an H1N1 virus (A/California/04/2009) were used for ADCC reporter assays; the sequence of the chimeric cH6/1 protein can be found in Appendix A. ADCC Jurkat effector cells expressing human FcɣRIIIa V158 were cultured in Roswell Park Memorial Institute (RPMI) 1640 media (Gibco, Paisley, UK) containing L-glutamine (Gibco, NY, USA), supplemented with 10% fetal bovine serum (FBS; HyClone, MA, USA), 100 μg/mL hygromycin (Invitrogen, CA, USA), 250 μg/mL antibiotic G-418 sulfate solution (Gibco, NY, USA), 1 mM sodium pyruvate (Gibco, NY, USA) and 0.1 mM minimal essential medium (MEM) of non-essential amino acids (Gibco, NY, USA). For enzyme-linked immunosorbent assays (ELISAs), the recombinant proteins cH6/1 (described above) and the mini-HA #4900 [18], which consists of a stabilized trimer of the stalk-domain from an H1N1 virus (A/Brisbane/59/2007), were used. To assess the neutralization capacity of the stalk-specific antibodies, a reassortant virus derived from an H1N1 virus (A/Puerto Rico/8/34) carrying the chimeric HA cH6/1 and the neuraminidase (NA) from an H12N5 virus (A/mallard/Sweden/86/2003) was used. The virus was grown in 8-day-old embryonated eggs (Charles River Laboratories, CT, USA) at 37 °C for 48 h. Human serum samples were obtained from a commercial vendor (110 samples). After testing, full units (volume~400 mL per donor) were purchased for the ten samples with highest reactivity to the HA stalk.

**Direct ELISA.** Antibodies in human serum were measured as described before [32]. In brief, ultra-high binding polystyrene 96-well plates (Immulon 4HBX; Thermo Scientific, PA, USA) were coated with 100 μL/well of recombinant protein in phosphate-buffered saline solution (PBS; pH 7.4; Gibco, NY, USA) at a concentration of 6 μg/mL for cH6/1 and 2 ug/mL for mini-HA #4900. Plates were incubated at 4 °C overnight, then washed 3 times with PBS containing 0.1% Tween 20 (PBS-T; Fisher Bioreagents, NJ, USA) using the plate washer system AquaMax 2000 (Molecular Devices, CA, USA). Blocking solution (220 μL/well) consisting of PBS-T, 3% goat serum (Gibco, OH, USA) and 0.5% non-fat dry milk (AmericanBio, MA, USA) was added to the plates, followed by incubation for 1–2 h. The serum was serially diluted (2-fold) from a 1:800 initial dilution for IgG and 1:100 for IgA. Samples were added to the plates (100 μL/well) and incubated at room temperature (RT) for 2 h. Plates were washed 3 times, and the specific secondary antibody (50 μL/well) was added at a 1:24,000 dilution. Goat Anti-Human IgG Fc specific horseradish peroxidase (HRP; Sigma, MO, USA) or Goat Anti-Human IgA α-chain specific HRP (Sigma, MO, USA) was used. After a 1 h incubation at RT, plates were washed 4 times and the substrate 3,3′,5,5′ tetramethylbenzidine (TMB, Bio-Rad, CA, USA) was added (100 μL/well). After a 30-min incubation, 50 μL/well of 4N H_2_SO_4_ solution (Thermo Fisher Scientific, MA, USA) was added. The optical density (OD) was measured at 450 nm using a Microplate Reader (Synergy H1, Biotek, VT, USA). Analysis was performed using Prism 7 software (GraphPad, San Diego, CA, USA), and values were reported as the area under the curve (AUC).

**Competition ELISA.** Coating using cH6/1 recombinant protein at 6 μg/mL and blocking were performed as for direct ELISAs. Plates were washed with PBS-T using the plate washer system AquaMax 2000 (Molecular Devices, CA, USA) as described above. Each lane on every 96-well plate was incubated for 2 h at RT with a specific serum sample to be tested at a 1:50 dilution in blocking solution (described above). Additionally, a non-competitor control plate was included, containing only blocking solution (described above). Two different setups for the competition ELISA assays were used according to the nature of the monoclonal antibody (mAb). For KB2 (mouse), 2-fold dilutions of the mAb (starting concentration of 0.08 μg/mL) were added (100 μL/well). Plates were incubated at RT for 2 h, washed 3 times, and incubated for 1 h with the secondary antibody Goat Anti-Mouse IgG (H&L) Antibody HRP (Rockland, PA, USA) at a 1:24,000 dilution (50 μL/well). MAb CR9114 (human) [7,33], was biotinylated using EZ-Link NHS-PEG4-Biotin (Thermo Scientific, IL, USA) and 2-fold dilutions of the mAb (starting concentration of 1 μg/mL) were added (100 μL/well) to every lane. Plates were incubated at RT for 2 h, washed 3 times, and incubated for 1 h with Pierce™ High Sensitivity Streptavidin-HRP (Thermo Scientific, IL, USA) at a 1:24,000 dilution (50 μL/well). Plates were washed 4 times, and substrate was added as described for direct ELISAs. Analysis was performed using Prism 7 software (GraphPad, San Diego, CA, USA), and values were reported as the percentage of competition between the antibodies contained in the serum samples and the mAbs.

**Microneutralization assay (MN).** Virus neutralization was assessed as previously described [34]. Briefly, MDCK cells maintained in DMEM (Gibco, NY, USA), supplemented with 10% FBS (HyClone, MA, USA) and Pen Strep (Gibco, NY, USA), were seeded in 96-well cell culture plates (Costar, DC, USA) and grown overnight at 37 °C with 5% CO_2_ to reach an approximate confluence of 80–90%. Serum samples were treated with a receptor-destroying enzyme (RDE, Denka Seiken, Japan) according to the manufacturer’s instructions and heat inactivated for 30 min at 56 °C. Serum samples were serially diluted (2-fold) from a 1:10 starting dilution in N-tosyl-L-phenylalanine chloromethyl ketone-treated trypsin-containing Ultra-MDCK medium (Lonza Bioscience, Belgium) and incubated for 1 h at room temperature with 100 times the 50% tissue culture infective dose (TCID_50_) of cH6/1N5 virus, to allow binding of the antibodies to the virus. MDCK cell-medium was removed, cells were washed with PBS, 100 μL/well of the serum-virus mixture was added and plates were incubated at 37 °C. After an incubation period of 1 h, the serum-virus mixture was removed, cells were washed with PBS, and replaced with 100 μL/well of diluted serum at the previous concentration. Infection was let to proceed for 48 h. Supernatants were collected and used to perform hemagglutination assay using chicken red blood cells (concentration: 0.5%) as described before [35]. Data were analyzed using Prism 7 software (GraphPad, San Diego, CA, USA), and values were reported as microneutralization titers.

**Antibody-Dependent Cellular Cytotoxicity assay (ADCC).** Evaluation of effector functions of antibodies was performed using a commercial ADCC reporter kit according to the manufacturer’s instructions (Promega, WI, USA). Briefly, cH6/1-MDCK cells were seeded in 96-well white flat bottom plates (Costar, ME, USA) at 3 × 10^4^ cells/well and plates were incubated overnight at 37 °C with 5% CO_2_. Serum samples were serially diluted (3-fold) starting from a 1:50 dilution in assay buffer consisting of RPMI 1640 medium supplemented with 0.5% low IgG FBS (Promega, WI, USA). Cell-growth medium was removed from cH6/1-MDCK cells and monolayers were washed with PBS (Gibco, NY, USA), followed by the addition of 25 μL/well of assay buffer and 25 μL/well of serum dilutions. Effector cells were thawed, washed and resuspended in assay buffer, and 7.5 × 10^4^ cells/well were added to each well in a volume of 25 μL. Plates were incubated at 37 °C with 5% CO_2_ for 6 h. Bio-Glo Luciferase Assay Reagent (Promega, WI, USA) was added (75 μL/well) and luminescence was measured using a Microplate Reader (Synergy H1, Biotek, VT, USA). Data were analyzed using Prism 7 software (GraphPad, San Diego, CA, USA), and values were reported as AUC.

**Pilot collaborative study and statistical analysis.** Eight laboratories from six countries participated in the study (Table 1). A sample panel (Appendix A) consisting of a total of twelve blinded samples was shipped to participating laboratories; the panel consisted of a pooled serum (candidate standard) in duplicate (samples 6 and 10) and ten individual samples with varying levels of anti-stalk reactivity (high, intermediate and low), selected from the 110 tested serum samples; all samples were blinded to participating laboratories. Participants were requested to test all samples for anti-stalk antibodies using any assay(s) of their choosing, with a minimum of three independent tests per sample and laboratory, and to record their results on a results template, supplied by the National Institute for Biological Standards and Control (NIBSC). Results were submitted to NIBSC, where ED50s (the 50% effective dilution corresponding to a half-maximal assay response) were calculated for all binding assays (including those where only AUC was reported by the participant) by NIBSC’s biostatisticians based on the submitted raw data where possible (one lab, for which independent calculation of ED50s was not possible, is indicated by an asterisk (*) in Appendix A). Analysis was performed with a four-parameter logistic (sigmoid curve) model using the R package ‘drc’ [36] and a log_10_ transformation of the assay readout in all laboratories. In 1 case (laboratory 12), 1 × 10^6^ was added to the assay readout value prior to log transformation and this was used as the assay response to calculate the sample ED50. Relative potencies were calculated by dividing the sample ED50 estimate by the corresponding assay ED50 estimate for the candidate standard sample 6. ED50 and potency estimates were combined as geometric means (GM) for each laboratory, and these laboratory means were used to calculate overall geometric means and overall median estimates for each sample. Variability between laboratories was expressed using geometric coefficients of variation (GCV = [10^s^ − 1] × 100% where s is the standard deviation of the log_10_ transformed estimates). The extent of deviation of individual laboratory estimates from study consensus values was expressed as the fold-change in the laboratory GM from the overall study median estimate for that sample.

## 3. Results

### 3.1. Generation of a Pooled Serum Containing High Levels of Stalk-Specific Antibodies

The production of a standard serum typically involves the collection of samples containing high levels of antibodies against the pathogen/molecule of interest. Several studies have demonstrated that some individuals possess higher levels of antibodies directed to the HA stalk of influenza viruses (likely induced by recent natural infection), and that these antibodies increase over time due to multiple exposures with influenza virus strains that are antigenically related [37,38]. Therefore, we decided to generate a standard serum containing high levels of stalk-specific antibodies by screening serum samples from healthy donors using a cH6/1 ELISA assay, which would allow us to detect antibodies directed specifically towards the stalk of group 1 HA influenza viruses (see workflow in Figure 1). Humans are naïve to the exotic avian H6 head domain, hence an undetectable amount of anti-head antibodies is present in human serum samples [39]. Anti-stalk antibodies measured using this chimeric protein have been shown to be an independent correlate of protection in humans [25]. Likewise, the cH6/1 construct, along with other chimeric constructs, has been used to assess stalk-specific antibodies in clinical trials for novel universal influenza vaccine candidates [27] (Nachbagauer et al., Nat Med. in press). Samples from a commercial vendor (*n* = 110) were screened for stalk-specific IgG titers (Figure 2A). The 10 samples with the highest reactivity were selected, and the full units from these donors were obtained (≈400 mL/sample). The full units were re-tested in the cH6/1 ELISA assay, to corroborate the presence of medium to high antibody titers against the stalk (Figure 2B). The 10 samples were mixed in equal proportions to generate a “pooled serum” that would comprise the model standard serum to be evaluated in this study, which exhibited high levels of IgG stalk-specific antibodies and relatively high levels of IgA stalk-specific antibodies (Figure 2C,D). Moreover, in order to characterize the properties of the antibodies contained in the serum samples, the samples were subjected to a panel of different assays that reflect the variety of tools currently available to detect group 1 stalk-reactive antibodies (Figure 1).

### 3.2. Characterization of Stalk-Specific Antibodies Contained in Serum Samples

Different assays are used to measure and characterize stalk-reactive antibodies in basic and clinical research settings. These include: binding assays such as ELISA [27,32] and bio-layer interferometry (BLI) [40]; assays to assess the neutralizing capacity of antibodies, such as the MN assay [41,42] and plaque reduction assay [40]; and tests to characterize the effector functions of antibodies, including ADCP and ADCC [16,33]. Here, we characterized the properties of stalk-specific antibodies contained in serum samples from healthy donors, including the samples that comprise the candidate standard serum, by a panel of different assays. Measurement of stalk-specific IgG by ELISA against the trimeric headless construct #4900 (mini-HA), which is recognized by a panel of different monoclonal antibodies directed against the group 1 HA stalk [18], allowed us to detect samples with variable antibody levels (Figure 3A). Moreover, comparison between IgG antibody levels against the chimeric protein cH6/1 and those against the mini-HA #4900 showed a strong and significant correlation (Figure 3B, Pearson r^2^: 0.7879; *p* (two-tailed): < 0.0001), which corroborates the presence and specificity of stalk-specific antibodies contained in the serum samples. To assess the neutralization capacity of the stalk-specific antibodies contained in the serum samples, we performed MN assays using the recombinant HA chimeric virus cH6/1N5, which allowed us to measure virus neutralization based on stalk-reactive antibodies. We found samples with variable levels of neutralization (Figure 3C), and a negligible correlation between the neutralization titers and the antibody levels measured in the cH6/1 ELISA was observed (Figure 3D, Pearson r^2^: 0.05219; *p* (two-tailed): 0.059); this was as expected because only subsets of binding antibodies, which vary between individuals, display neutralizing activity. Moreover, despite binding, differences in the in vitro neutralization activity of stalk and head antibodies are observed [43] and neutralization by stalk antibodies substantially depends on their effector functions such as ADCC activity [16], which play a role in vivo and are not detected in the in vitro microneutralization assay. Selected samples from low, intermediate and high responders in the cH6/1 ELISA were tested in an ADCC assay using a cell line stably expressing the chimeric HA cH6/1. Due to the low number of samples tested, correlation analysis could not be performed, however a positive association between the cH6/1 antibody titers and the ADCC activity was observed (Figure 3E), indicating that the stalk-reactive antibodies present in the serum samples possess effector functions, which may be important for in vivo protection [16]. Finally, in order to assess whether antibodies contained in the serum samples bind to some of the conserved epitopes in the stalk domain of the HA, we performed competition ELISAs using the widely characterized stalk-reactive monoclonal antibodies KB2 (mouse) and CR9114 (human) [7,33,44]. Using samples from high responders in the cH6/1 ELISA, we were able to detect a high percentage of competition (above 30% in most cases) between the monoclonal antibodies and the stalk-reactive antibodies contained in the serum samples (Figure 3F). In summary, these results confirmed the presence and specificity of stalk reactive antibodies in serum from healthy donors, using an array of different assays available for assessment of stalk-specific antibody responses.

### 3.3. Testing of the Candidate Serum Standard in a Collaborative Study

The establishment of an international standard would require testing in multiple laboratories worldwide. Here, we conducted a pilot collaborative study to assess the potential of the candidate standard (pooled serum) to harmonize results from multiple assays from different laboratories. A sample panel consisting of the pooled serum and samples with varying levels of anti-stalk reactivity, selected from the 110 tested serum samples, was sent to the participating laboratories (Appendix A, Table 1); all samples were blinded to participating laboratories. Participants were requested to test all samples for anti-stalk antibodies using any assay(s) of their choice, with a minimum of three independent tests per sample and laboratory. An array of different assays was used in the laboratories to assess binding and functional properties of stalk-reactive antibodies. Only binding assays and neutralization assays, for which results from at least two different laboratories were available, are reported here (Appendix A).

Participants were instructed to record their results on a specific template to allow a common analysis of all data at NIBSC. Not all participating laboratories returned data by the study deadline, while some laboratories supplied more than one dataset; laboratories were assigned random numbers, not related to the order of laboratories as shown in Table 1. Where the result from a single assay run caused the range of ED50 estimates to exceed eight-fold for a sample within a laboratory, the result was considered to be an outlier and was excluded from further analysis; the small number of cases where this occurred are indicated in Appendix A. Geometric mean ED50 estimates and geometric mean potency estimates relative to candidate standard sample 6 are shown in Figure 4 and Appendix A. Samples 3 and 11 gave low relative potencies with GM < 0.05, below the limit of detection in some laboratories, and were therefore excluded from subsequent analysis. A reduction in between-laboratory %GCV when expressing titers as relative potencies was observed for all the other samples.

The extent of deviation from study consensus values for individual laboratories was assessed by calculating the fold-change of laboratory GM from the overall median estimate for each sample. This was calculated for both ED50 and relative potency estimates (Table 2 and Table 3, Figure 5). Values closer to 1.0 indicate better agreement of a laboratory’s result with the overall study median. Good inter-laboratory agreement (harmonization) when titers were normalized to the pooled serum (sample 6) was evident for labs 4, 7, 8, 9, 12a and 12b: 89–100% of potencies were within four-fold of the overall study sample median. In contrast, none of ED50 estimates for labs 9 and 12a were within this range without normalization (Figure 5A). Poorer agreement following normalization was observed for lab 11, while no change was evident for lab 10.

Intra-laboratory variability (test repeatability within each laboratory) was assessed as ratios of the maximum and minimum ED50s or relative potencies for each sample in each laboratory (Figure 6 and Appendix A). Normalization using sample 6 had no noticeable impact on intra-laboratory variability. The candidate standard was included twice in the blinded sample panel (samples 6 and 10), which also allows assessment of intra-laboratory variability; the relative potency of sample 10 should be close to 1, which was the case for most laboratories, but not for laboratory 11 (Appendix A).

Only two laboratories returned data from virus neutralization assays, limiting the statistical analysis that could be performed. The differences in GM endpoint titer estimates and the GM relative potency estimates between the two labs for each sample are illustrated by the max:min ratios shown in Table 4 and Table 5. The use of normalization relative to sample 6 reduced the max:min ratio for most samples (10 out of 11), indicating the potential for harmonization of results by the use of a standard.

## 4. Discussion

Multiple studies underline the importance of HA stalk-specific antibodies in the prevention and outcome of influenza virus infections [22,23,24]. Indeed, stalk-specific antibodies have been pointed out as independent correlates of protection [25]. Therefore, qualitatively and quantitatively measuring these particular types of antibodies and comparison of laboratory results among different research groups are essential. Here, we generated, characterized, and tested a candidate serum standard for stalk-reactive antibodies in humans in an international pilot collaborative study. The candidate standard exhibited high levels of stalk-reactive antibodies, a high neutralization titer, and displayed strong antibody-effector functions such as high levels of ADCC activity.

Results obtained from the international pilot study support the concept that implementation of a standard would improve the harmonization of results from different laboratories. Normalization of results to the standard improved inter-laboratory variability of stalk-specific antibody levels. The generation of a standard based on human sera ensures that the matrix of the reagent is compatible with analysis of stalk-reactive antibodies in human serum samples, providing commutability with samples of clinical importance, such as samples from clinical trials of prophylactic approaches and in diagnostic settings. Moreover, antibodies in human serum samples stored at −20 °C are well preserved for prolonged periods of time [45]; therefore, we anticipate that a future international standard, generated analogously to the one described here, and stored at low temperature after lyophilization, will be stable [46]. Even though normalization to the candidate standard did not improve agreement to the overall median for all laboratories or all samples, this observation is not unusual [47,48]. Reduced harmonization may have been due to multiple causes, including inexperience of a laboratory with a particular assay, variability of reagents used, or systematic differences between tests of the standard and the samples. Moreover, testing results exhibited similar patterns of inter- and intra- laboratory variability as seen with other standards [46,49].

In summary, these results suggest that the use of a standard has the potential to facilitate the comparison of experiments to measure stalk-specific antibodies conducted in different laboratories and to increase confidence in results. We therefore conclude that the generation of an international standard, based on the results of this Phase 1 project, and using the same pool of high-titer samples that was tested in this study, is worthwhile, and will be of use in the development and assessment of vaccine candidates targeting the HA stalk domain.

## Figures and Tables

**Figure 1 vaccines-08-00666-f001:**
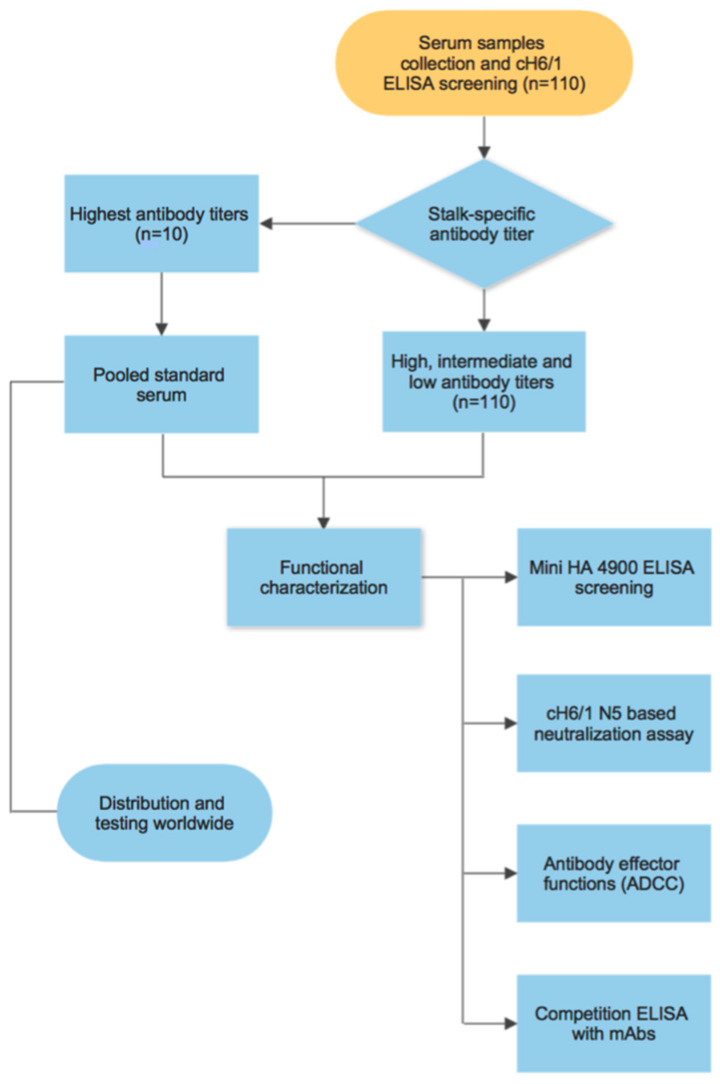
Workflow of the generation and characterization of a pooled serum as a candidate standard to measure influenza virus hemagglutinin stalk-reactive antibodies.

**Figure 2 vaccines-08-00666-f002:**
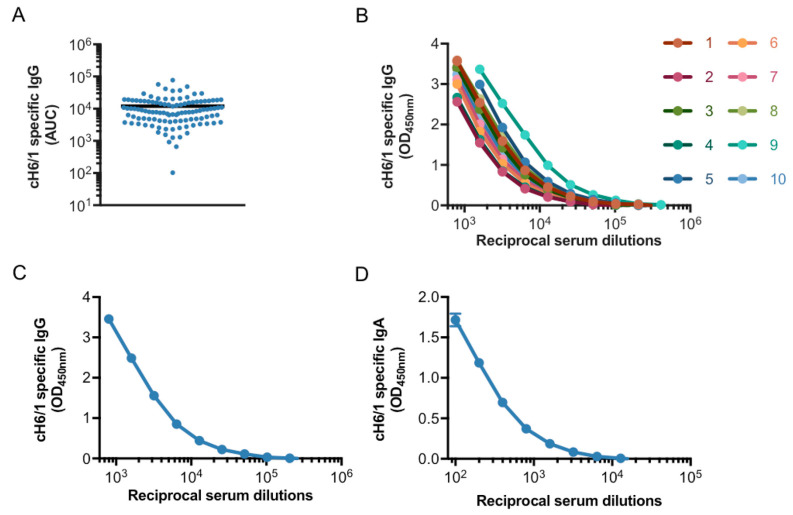
**Testing and selection of human serum samples with high levels of stalk-specific antibodies.** Samples of human sera were obtained from a commercial vendor (*n* = 110) and screened for stalk-specific IgG antibodies using a cH6/1-based enzyme-linked immunosorbent assay (ELISA) (**A**). The 10 samples with the highest IgG titers were selected, and the full units were obtained. The full units were re-tested for cH6/1-specific IgG titers (**B**). A pooled serum (PS) consisting of equal amounts of serum from each of the 10 full units was generated. The PS exhibited high levels of stalk-specific IgG (**C**) and IgA (**D**) antibodies. Dots in A represent individual values of the Area Under the Curve (AUC) from every serum sample, the arithmetic mean of all values is represented by a black horizontal line. Specific Optical Density (OD) for each of the serum dilutions is shown in (**A**–**C**).

**Figure 3 vaccines-08-00666-f003:**
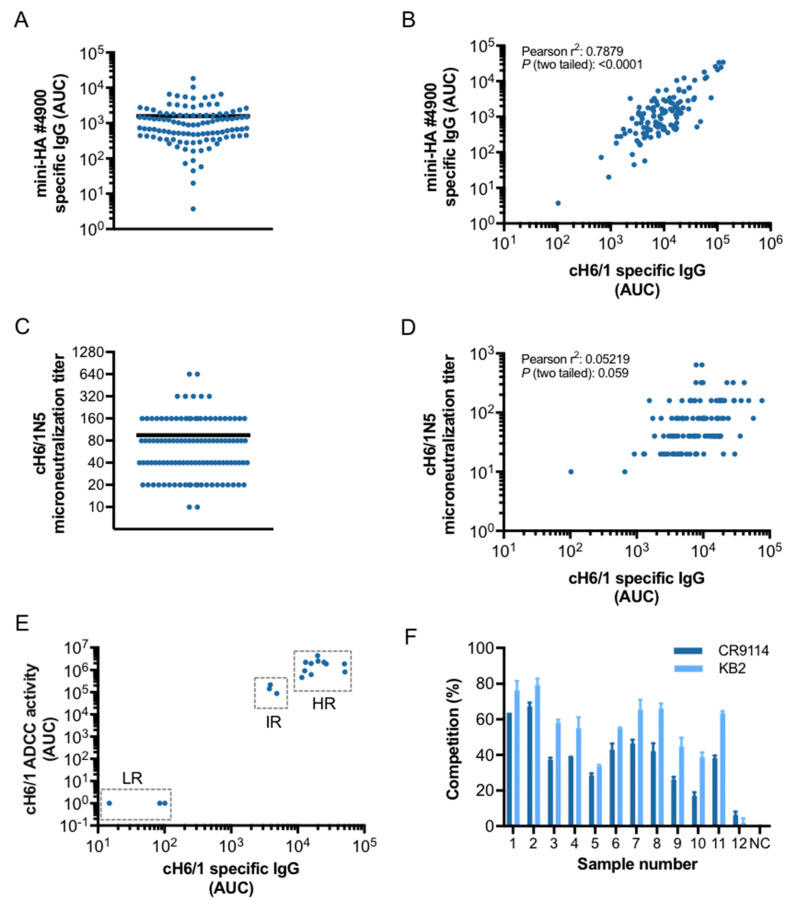
**Characterization of stalk-specific antibodies contained in human serum samples.** Human serum samples (*n* = 110) were screened for stalk-specific antibodies using a #4900 mini HA-based ELISA (**A**). The correlation between cH6/1-specific IgG and #4900 mini HA-specific IgG antibody levels is shown (**B**). Serum samples (*n* = 110) were subjected to a cH6/1N5-based neutralization assay. Microneutralization titers obtained are presented (**C**). The correlation between cH6/1-specific IgG and microneutralization titers is shown (**D**). Samples from low, intermediate and high responders in the cH6/1 ELISA were tested in an ADCC commercial assay (*n* = 17). Association of the cH6/1-specific IgG levels with the effector functions of the antibodies measured in the ADCC assay is shown (**E**). Competition of the antibodies contained in the serum samples and the monoclonal antibodies KB2 (mouse) and CR9114 (human) was determined and presented as percentage of competition (**F**). Dots in (**A**,**B**,**D**,**E**), represent individual values of area under the curve (AUC) from every serum sample. The arithmetic mean of all values is represented by a black horizontal line. Pearson correlation coefficient (r^2^) and *p*-value are shown in (**B**,**D**).

**Figure 4 vaccines-08-00666-f004:**
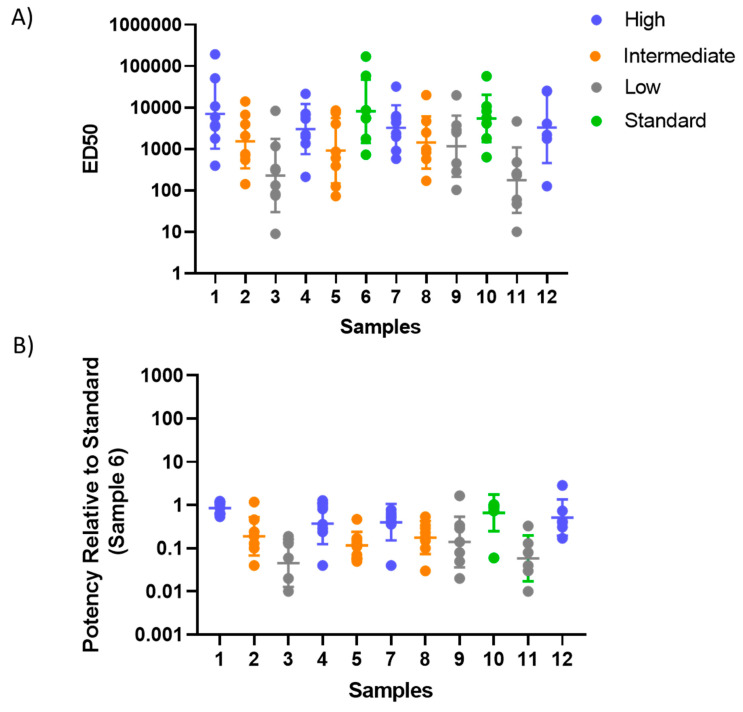
**Geometric mean ED50 (50% effective dilution corresponding to a half-maximal assay response) and relative potency estimates**. ED50 (**A**) and relative potency (**B**) estimates are shown for all samples (*n* = 11) and laboratories (*n* = 8). Each dot represents a geometric mean estimate for one sample and one laboratory. The same data are shown in Appendix A and Appendix A.

**Figure 5 vaccines-08-00666-f005:**
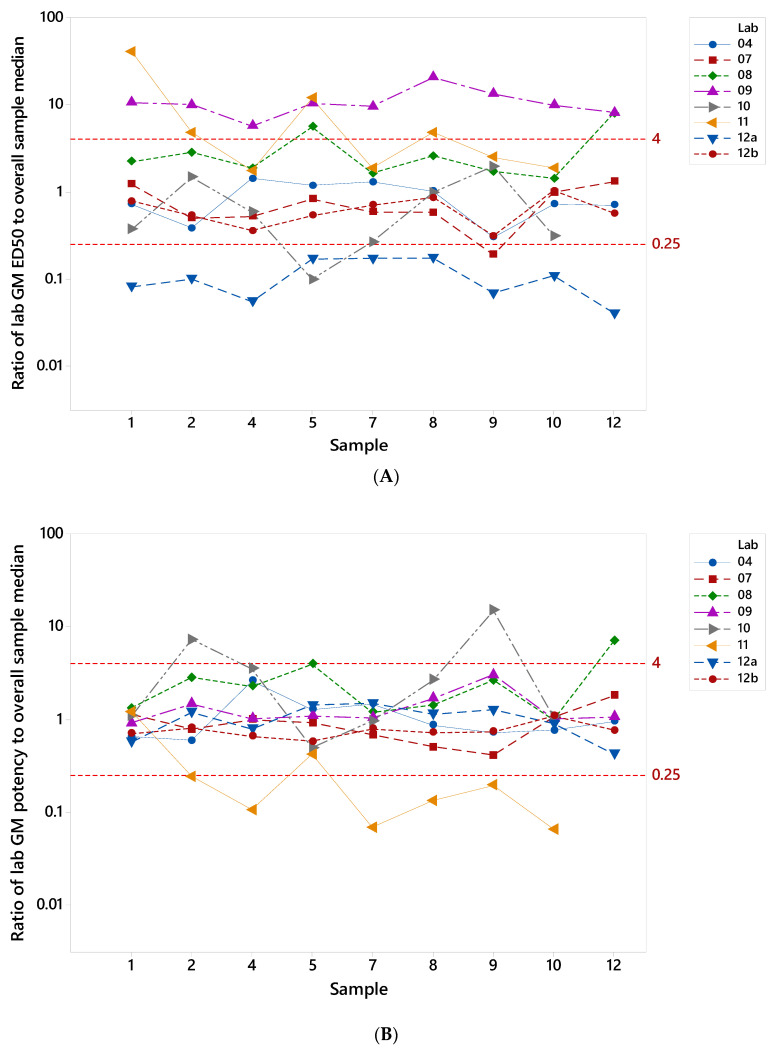
**Inter-laboratory variability in ED50 and relative potency estimates**. Individual points show the ratio of laboratory geometric mean ED50 estimates (**A**) and relative potency estimates (**B**) to the study median ED50 estimate for that sample; range of 0.25–4 is shown to indicate points that are no more than 4-fold different from the study median.

**Figure 6 vaccines-08-00666-f006:**
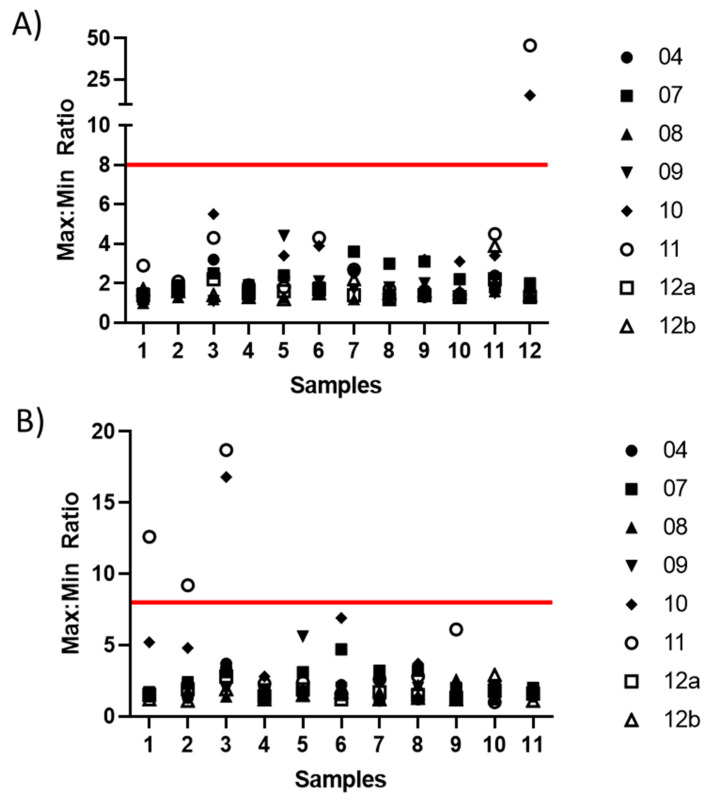
**Intra-laboratory variability in ED50 and relative potency estimates.** Individual points show the max:min ratio of laboratory ED50 estimates (**A**) and relative potency estimates (**B**) for each sample and laboratory; the red line marks a max:min ratio of 8-fold.

**Table 1 vaccines-08-00666-t001:** Laboratories participating in pilot collaborative study.

Institution	Name	Country
Janssen Vaccines & Prevention	Boerries Brandenburg	The Netherlands
Centers for Disease Control and Prevention	Min Levine	United States
University of Bergen	Fan Zhou	Norway
Icahn School of Medicine at Mount Sinai	Adolfo García-Sastre & Teresa Aydillo-Gomez	United States
Vismederi Research Srl.	Alessandro Manenti	Italy
National Institutes of Health	Barney Graham	United States
University of Hong Kong	Sophie Valkenburg	China SAR
National Institute of Biological Standards & Control	Lethia Charles & Othmar Engelhardt	United Kingdom

**Table 2 vaccines-08-00666-t002:** Inter-lab variability: Fold-change of laboratory geometric mean ED50 estimates from the overall study median ED50 estimate for each sample.

Sample	Laboratory
4	7	8	9	10	11	12a	12b
**1**	1.39	1.22	2.22	10.46	2.70	40.18	12.30	1.29
**2**	2.60	2.02	2.81	9.93	1.47	4.78	9.96	1.88
**4**	1.41	1.92	1.87	5.65	1.70	1.74	18.06	2.80
**5**	1.18	1.21	5.55	10.32	10.08	11.73	5.90	1.87
**7**	1.30	1.71	1.63	9.54	3.75	1.86	5.84	1.42
**8**	1.02	1.72	2.57	20.47	1.02	4.78	5.76	1.17
**9**	3.31	5.24	1.69	13.16	1.94	2.49	14.52	3.23
**10**	1.39	1.02	1.42	9.85	3.21	1.86	9.18	1.02
**12**	1.43	1.30	7.88	8.11	N/A	N/A	24.64	1.77
**% < 2**	**78%**	**78%**	44%	**0%**	**50%**	**34%**	**0%**	**78%**
**% < 4**	**100%**	**89%**	**78%**	0%	**88%**	**50%**	**0%**	**100%**
**% < 8**	**100%**	**100%**	**100%**	**11%**	**88%**	**75%**	**25%**	**100%**

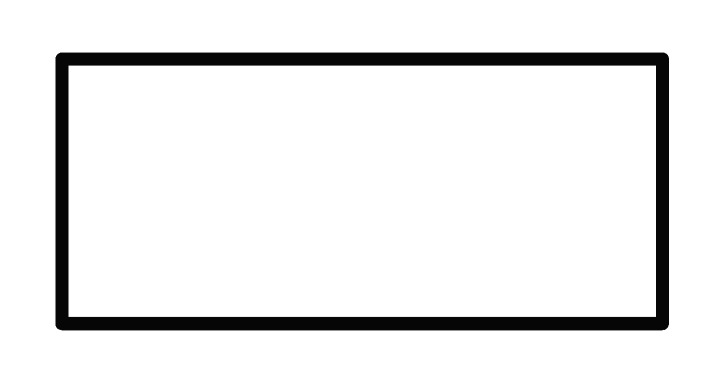
: X < 2; 
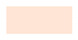
: 2 < X < 4; 
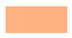
: 4 < X < 8; 

: X > 8.

**Table 3 vaccines-08-00666-t003:** Fold-change of laboratory geometric mean potency estimates from the overall median potency estimate for each sample.

Sample	Laboratory
4	7	8	9	10	11	12a	12b
**1**	1.53	1.14	1.32	1.09	1.08	1.21	1.75	1.43
**2**	1.70	1.28	2.82	1.47	7.23	4.13	1.19	* 1.23
**4**	2.61	1.01	2.26	1.01	3.50	9.39	1.26	* 1.52
**5**	1.28	1.09	3.94	1.08	2.03	2.38	1.42	* 1.73
**7**	1.46	1.48	1.20	1.04	1.04	14.49	1.50	* 1.27
**8**	1.16	1.98	1.42	1.68	2.67	7.48	1.14	1.39
**9**	1.39	2.44	2.63	3.02	14.83	5.12	1.27	* 1.36
**10**	1.31	1.07	1.01	1.01	1.07	15.34	1.12	1.08
**12**	1.06	1.80	6.96	1.06	N/A	N/A	2.36	* 1.32
**% < 2**	**89%**	**89%**	**44%**	**89%**	**34%**	**13%**	**89%**	**100%**
**% < 4**	**100%**	**100%**	**89%**	**100%**	**75%**	**25%**	**100%**	**100%**
**% < 8**	**100%**	**100%**	**100%**	**100%**	**88%**	**63%**	**100%**	**100%**

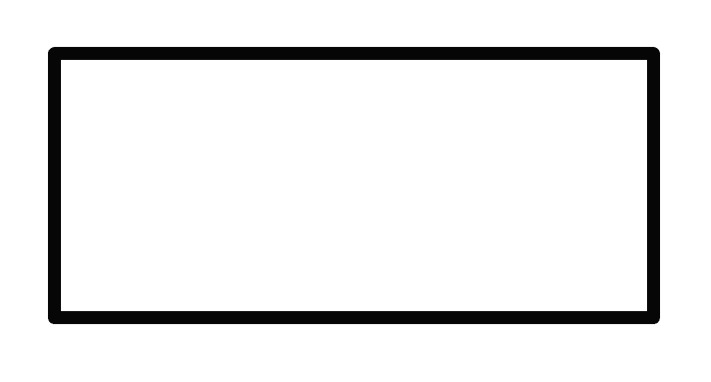
: X < 2; 
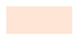
: 2 < X < 4; 
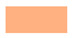
: 4 < X < 8; 

: X > 8. * max:min ratio reduced when potency expressed relative to sample 6.

**Table 4 vaccines-08-00666-t004:** Geometric mean endpoint readout estimates—Virus Neutralization assays.

Sample	Laboratory		GM	Ratio	Max:Min Ratio
	04	10			
1	84.8	67.3	75.5	1.26	1.26
2	95.3	40	61.7	2.38	2.38
3	80	33.6	51.9	2.38	2.38
4	80	67.3	73.4	1.19	1.19
5	40	40	40	1.00	1.00
7	160	80	113.1	2.00	2.00
8	89.9	28.3	50.4	3.18	3.18
9	80	28.3	47.6	2.83	2.83
10	160	47.6	87.2	3.36	3.36
11	20	10	14.1	2.00	2.00
12	80	40	56.6	2.00	2.00

Shading shows ratios ≥ 2.00.

**Table 5 vaccines-08-00666-t005:** Geometric mean potency estimates relative to Sample 6—Virus Neutralization assays.

Sample	Laboratory		GM	Ratio	Max:Min Ratio
	04	10			
1	0.53	0.59	0.56	0.89	* 1.11
2	0.60	0.35	0.46	1.68	* 1.71
3	0.50	0.30	0.39	1.68	* 1.67
4	0.50	0.59	0.55	0.84	* 1.18
5	0.25	0.35	0.30	0.71	1.40
7	1.00	0.71	0.84	1.41	* 1.41
8	0.56	0.25	0.37	2.25	* 2.24
9	0.50	0.25	0.35	2.00	* 2.00
10	1.00	0.42	0.65	2.38	* 2.38
11	0.13	0.08	0.10	1.59	* 1.63
12	0.50	0.35	0.42	1.41	* 1.43

Shading shows ratios ≥ 2.00. * max:min ratio reduced when potency expressed relative to sample 6.

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
