# Peer review of "Development and Assessment of a Pooled Serum as Candidate Standard to Measure Influenza A Virus Group 1 Hemagglutinin Stalk-Reactive Antibodies"

_vaccines, 2020, doi:10.3390/vaccines8040666_

Round 1

Reviewer 1 Report

In this manuscript by Juan Manuel Carreno et al., the authors set out to harmonize the assays to evaluate the antibodies and humoral response directed against the stalk of influenza A virus hemagglutinin. To that end they screened 110 serum samples from healthy individuals using an ELISA that measures stalk-reactive antibodies. They also evaluated their activity through virus neutralization and antibody-dependent cell-mediated cytotoxicity. They constituted a pooled serum (PS) that was assayed independently in several collaborating laboratories, and that could constitute a universal serum standard to test the antibodies against the HA stalk. Indeed there is a need for reference standards to calibrate the assays and allow comparison of biological measurements.

Main findings

  • The authors screened 110 serum samples by a cH6/1 ELISA assay (specific toward the stalk of group 1 hemagglutinins). The 10 best samples were pooled (Pooled Serum, PS).
  • The IgG antibody levels against the chimeric protein cH6/1 were strongly correlated to those against the mini-HA (headless construct #4900), and showed only a modest correlation with the virus-neutralization activity. The sera with high cH6/1 antibody levels also showed an ADCC activity (antibody-dependent cellular cytotoxicity) towards a cell line expressing the chimeric cH6/1, as well as high percentages of competition against stalk-specific monoclonal antibodies.
  • Selected samples (among the 110 sera), along with the PS, were evaluated in eight collaborative laboratories, each using the assay(s) of its choice.
  • All the data were collected, then analysed by the leading laboratory, who expressed the assay results as ED50 estimates and potency estimates relative to the candidate standard (PS=sample 6)
  • As an attempt towards harmonization, the extent of deviation from consensus values was assessed, being expressed as the fold-change from the overall median estimate (good harmonization would translate as fold-change values close to 1.0)

The study is very interesting, and focuses on a very important issue. The manuscript is well written. The data are probably not as clean as one could expect, but this emphasizes the need for standard reagents and standard methods. There are some points that need to be addressed by the authors in order to improve the manuscript.

Major remarks

Figure 2A. The geometric mean should be represented by a horizontal line, not a rectangle (and what exactly are the error bars, SD or geometric SD?, geometric SD should be preferred). The error bars are not necessary, since all the points are displayed.

Same remarks also for Figures 3A and 3C, and suppl. Figure 1.

Suppl. Figure 1. The y-scales should be adapted to the distribution of the points (the y-scale should begin at about 103, 102 and 105 in A, B, and D, respectively).

Figure 4 (A and B). The y-scale should be adapted to the distribution of points (start at 1 for A, and for B the y-scale should be [0.001-10]).  

Lines 258-59. Why “as expected, a negligible correlation”? The authors should explain why this is expected (here or in the Discussion).  

Perhaps it should be useful to know the “test repeatability” for any given participating lab (at least after normalization relative to sample 6), since each lab conducted a minimum of three independent tests per sample. This is perhaps the meaning of suppl. table 6 (since most labs used a single assay), but this is not clear. 

In Suppl. Figure 2, it would be useful to connect all the points belonging to each lab (for instance, connect the GM values for samples 1-12 for lab #4, the same for lab #7, etc). This would facilitate the identification of the points (10 and 11 are difficult to distinguish, as are 04 and 12b), and may also add some information: if the lab-specific lines do not cross each other, the ranking of the samples is the same across the labs). Further, the points in suppl. Figure 2 do not seem to match the values in suppl. Table 5.  

Minor remarks

Line 323 ….without normalization (Figure 5A).

Suppl. table 2, line 3, column 4: what is protein cH9/1HA?

Reviewer 2 Report

To measure influenza hemagglutinin stalk-based immunity is necessary for development of effective universal influenza vaccines. There is a need for standardized measurement of such stalk-based immunity. In this study, serum samples from heathy individuals were screened for stalk-reactive antibodies for binding, neutralization and ADCC. A "pooled serum" of mixture of selected serum samples was generated with high levels of stalk-reactive antibodies. This pooled serum was tested as standard along with serum samples of varying stalk antibody titers by multiple collaborators worldwide. Experimental results for binding and neutralization were statistically analyzed with some deviations for different labs. The idea of standard serum sample is interesting, and the material is suitable for standardized experiments. 

Some minor suggestions:

Page 3, line 109-111. The exact amino acid composition of cH6/1 should be provided here.

Page 4, 193-194. A detailed explanation of ED50 would be helpful.

Page 5, line 215-219. An explanation should be given why antibodies binding to cH6/1 would be stalk-binding antibodies. A binding test to H6 head only would be beneficial as a negative control for stalk-binding antibodies.
